# Duodenum Intestine-Chip for preclinical drug assessment in a human relevant model

Magdalena Kasendra[1†]*, Raymond Luc[1†], Jianyi Yin[2], Dimitris V Manatakis[1], Gauri Kulkarni[1], Carolina Lucchesi[1], Josiah Sliz[1], Athanasia Apostolou[1,3], Laxmi Sunuwar[2], Jenifer Obrigewitch[1], Kyung-Jin Jang[1], Geraldine A Hamilton[1], Mark Donowitz[2], Katia Karalis[1]

[1]Emulate Inc, Boston, United States; [2]Department of Medicine, Division of Gastroenterology, Johns Hopkins University School of Medicine, Baltimore, United States; [3]Graduate Program, Department of Medicine, National and Kapodistrian University of Athens, Athens, Greece

**Abstract** Induction of intestinal drug metabolizing enzymes can complicate the development of new drugs, owing to the potential to cause drug-drug interactions (DDIs) leading to changes in pharmacokinetics, safety and efficacy. The development of a human-relevant model of the adult intestine that accurately predicts CYP450 induction could help address this challenge as species differences preclude extrapolation from animals. Here, we combined organoids and Organs-on-Chips technology to create a human Duodenum Intestine-Chip that emulates intestinal tissue architecture and functions, that are relevant for the study of drug transport, metabolism, and DDI. Duodenum Intestine-Chip demonstrates the polarized cell architecture, intestinal barrier function, presence of specialized cell subpopulations, and *in vivo* relevant expression, localization, and function of major intestinal drug transporters. Notably, in comparison to Caco-2, it displays improved CYP3A4 expression and induction capability. This model could enable improved *in vitro* to *in vivo* extrapolation for better predictions of human pharmacokinetics and risk of DDIs.

*For correspondence:
magdalena.kasendra@gmail.com

†These authors contributed equally to this work

## Introduction

Low bioavailability and pharmacokinetics caused by drug-drug interactions of orally administered drugs represents a significant challenge in the modern drug development. High affinity of certain drugs for cellular transporters combined with the extensive activity of metabolic enzymes present in the human intestine are the main factors limiting drug bioavailability (*Dietrich, 2003*; *Thummel, 2007*; *Shugarts and Benet, 2009*; *Peters et al., 2016*). After decades of research, the hepatic drug clearance is well-understood and relatively well predicted by pre-clinical models, while the accurate prediction of the first-pass extraction of xenobiotics in human intestinal epithelium still remains elusive. This is due to a number of confounding factors that affect oral drug absorption including the properties of the compound (solubility, permeability), physiology of the intestinal tract (transit time, blood flow), and patient phenotype (including age, gender, polymorphism in drug metabolizing enzymes, disease states) (*Pang, 2003*). Species differences in the isoforms, regional abundances, differences in substrate specificity of drug metabolism enzymes (*Martignoni et al., 2006*; *Paine et al., 2006*; *Komura and Iwaki, 2011*) and transporters (*Tucker et al., 2012*; *Gröer et al., 2013*), and mechanism regulating transcriptional activation (*LeCluyse, 2001*; *Mackowiak et al., 2018*), precludes accurate extrapolation of the data from animal models to the clinic. In mice, for example, there are 34 cytochrome P450 (CYPs) in the major gene families involved in drug metabolism, that is the *CYP1A*, *CYP2C*, *CYP2D*, and *CYP3A* gene clusters, while in humans,

there are only eight (*Nelson et al., 2004*). Interestingly, three human enzymes, CYP2C9, CYP2D6, and CYP3A4, account for ~75% of all reactions, with CYP3A4 being the single most important human CYP450 accounting for ~45% of phase one drug metabolism in humans (*Guengerich, 2008*). In addition, the expression levels of many of the major human CYP450 enzymes and drug transporter (which determine levels and variability in drug exposure) are controlled by multiple transcription factors, primarily the xenosensors: constitutive androstane receptor (CAR), pregnane X receptor (PXR), and aryl hydrocarbon receptor (AhR). These nuclear receptors also exhibit marked species differences in their activation by drugs and exogenous chemicals (*Mackowiak et al., 2018*). For example, rifampicin and SR12813 are potent agonists for human PXR (hPXR) but not for mouse PXR (mPXR), whereas the potent mPXR agonist 5-pregnen-3β-ol-20-one-16α-carbonitrile (PCN) is a poor agonist for hPXR (*Kliewer et al., 1998*). On the other hand, 6-(4-chlorophenyl)imidazo[2,1-b][1,3]thiazole-5-carbaldehyde-O-(3,4-dichlorobenzyl)oxime (CITCO) is a strong agonist for human CAR (hCAR) but not mouse CAR (mCAR) (*Maglich et al., 2003*), while 1,4-bis-[2-(3,5-dichloropyridyloxy)] benzene,3,3′,5,5′-tetrachloro-1,4-bis(pyridyloxy)benzene (TCPOBOP) is more selective for mCAR than hCAR. Such species differences together with the complex interplay between drug metabolizing enzymes and drug transporters in the intestine and liver, as well as the overlap of substrate and inhibitor specificity (*Shi and Li, 2014*), make it difficult to predict human pharmacokinetics at the preclinical stage of drug development.

Numerous *in vitro* models have been developed and applied routinely for characterization and prediction of absorption, distribution, metabolism, and excretion (ADME) of potential drug candidates in humans. Among these is a Caco-2 monolayer culture on a transwell insert, which is one of the most widely used models across the pharmaceutical industry as an *in vitro* representation of the human small intestine. However, inherent limitations, such as lack of *in vivo* relevant three-dimensional (3D) cytoarchitecture, lack of appropriate ratio of cell populations, altered expression profiles of drug transporters and drug metabolizing enzymes, especially CYP450s, and aberrant CYP450-induction response, challenge the use of these model for predicting human responses in the clinic (*Sun et al., 2008*).

A promising alternative to conventional cell monolayer systems emerged with the establishment of the protocols for a generation of 3D intestinal organoids (or enteroids) from human biopsy specimens (*Sato et al., 2009*; *Sato et al., 2011*; *Eglen and Randle, 2015*; *Liu et al., 2016a*). Using these methods, organoids derived from all regions of the intestinal tract can be established (*Sato et al., 2011*; *Wang et al., 2015*) and applied into different areas of research including organ development, disease modeling, and regenerative medicine (*Fatehullah et al., 2016*). However, the characterization of the pharmacokinetic properties of this system, as well as the validation for its use in drug discovery and development, is still very limited (*Dekkers et al., 2013*; *van de Wetering et al., 2015*; *Zhang et al., 2017*; *Zhao et al., 2017*; *Vlachogiannis et al., 2018*). This could be due to the substantial technical challenges associated with the use of organoid technology for ADME applications. The 3D spherical architecture of the organoid restricts access to its lumen, which is crucial for assessing intestinal permeability or drug absorption. Indeed, exposure of the apical cell surface of intestinal organoid to compounds requires the use of time-consuming and labor-intensive procedures, such as microinjection. The presence of relatively thick gel of extracellular matrix (Matrigel) surrounding organoids, might limit drug penetration. While heterogeneity of organoids in terms of their size, shape, and viability, can also impede studies in ADME and adequate interpretation of the results (*Fatehullah et al., 2016*).

In addition, none of these models have fully recapitulated critical aspects of an organ microenvironment such as the presence of microvasculature, mechanical forces of fluid flow (shear stress) and peristalsis, all of which contribute to the complex and dynamic nature of *in vivo* tissue function (*Gayer and Basson, 2009*). For these reasons, there is an urgent need for new *in vitro* models able to accurately predict human ADME and to identify early the risk for DDIs mediated by intestinal CYP450s and drug transporters in the clinic.

We have recently developed a human Duodenum Intestine-Chip that combines healthy intestinal organoids with our Organs-on-Chips technology (*Kasendra et al., 2018*) to overcome the existing limitations of current systems. Here, we demonstrated that Duodenum Intestine-Chip provides a human-relevant model for the studies of drug transport, drug metabolism, and DDIs, which is much closer to the human tissue, compared to organoids, as supported by the comparison of RNA sequencing (RNA-seq) expression profiles. Presence of the mechanical forces, which we apply in this

system in order to recapitulate the blood flow and shear stress, showed to improve the formation of polarized cytoarchitecture and the appearance of intestinal microvilli on the apical cell surface. In addition, Duodenum Intestine-Chip supported successful maturation of all major intestinal epithelial cell types in the physiologically relevant ratios and demonstrated low paracellular permeability. Importantly for its use in pharmacokinetic studies, it showed closer to *in vivo* expression of drug uptake and efflux transporters, as compared to Intestine-Chip developed with Caco-2 cells. Additionally, we were able to show that the Duodenum Intestine-Chip exhibits the correct luminal localization and functional activity of MDR1 (P-gp), and supports high expression of the cytochrome CYP450 (CYP) 3A4 - close to the levels observed in the human duodenal tissue. Importantly, exposure of the Duodenum Intestine-Chip to known CYP450 inducers in humans, such as rifampicin and 1, 25-dihydroxyvitamin D3, resulted in significantly increased CYP3A4 mRNA, protein levels, and drug metabolism activity. Our results indicate that the organoid-derived intestinal cells can be combined with Organs-on-Chips technology to provide a robust and human-relevant system for preclinical assessment of CYP450-mediated metabolism, activity of drug transporters, and the potential risk of DDIs.

## Results

### Development of the adult Duodenum Intestine-Chip

We have previously developed a human primary Intestine-Chip, referred at the time as the 'Small Intestine-on-a-Chip', which combined the use of intestinal organoids isolated from pediatric donors and Organ-Chips (*Kasendra et al., 2018*) to recapitulate critical features of intestinal morphology and associated functions. Here, we sought to establish a Duodenum Intestine-Chip from adult organoid-derived cells to serve as a platform in the preclinical assessment of drug transport and metabolism. In brief, we first established organoid cultures (*Figure 1A*; top) from crypts isolated from endoscopic biopsies of three different healthy adult individuals, the organoids were then dissociated into fragments and seeded on the top of the Extracellular Matrix (ECM)-coated porous flexible polydimethylsiloxane (PDMS) membrane of the chips (*Figure 1B*; 3: indicates the epithelial tissue). Primary human intestinal microvascular endothelial cells (HIMECs, Cell Biologics), derived from the human small intestine (*Figure 1A*; bottom), were used to populate the other surface of the PDMS membrane in the vascular channel (*Figure 1B*; 4: indicates the endothelial cells). Next, the Duodenum Intestine-Chip was perfused continuously through the luminal and vascular compartment with fresh cell culture medium. Once the epithelial monolayers reached confluency they were subjected to cyclic mechanical strain (10% strain, 0.2 Hz) in order to emulate physiologically relevant forces of intestinal peristalsis.

Next, we assessed the effect of applied mechanical stimulation (in the form of fluid flow and stretch) on the phenotypic characteristics of the human primary intestinal cells grown on the chip. To this end, we used multiple endpoints including immunofluorescent staining for apical (villin) and basolateral (E-cadherin) cell surface markers and scanning electron microscopy (SEM) for the identification of apical microvilli. Exposure of the Duodenum Intestine-Chip to flow for 72 hr resulted in accelerated polarization of the epithelial cells and formation of apical microvilli (*Figure 1—figure supplement 1*), what is in line with our previous findings reported for the Caco-2 cells (*Kim et al., 2012*). We observed that culture of primary intestinal epithelial cells in the chips maintained under static conditions resulted in the formation of a monolayer of flat (14.8 ± 2.6 µm) squamous cells with poorly defined cell-cell junctions (*Figure 1—figure supplement 1A*) and sparsely distributed microvilli (*Figure 1—figure supplement 1B*). In contrast, cells cultured under flow (30 µl/hr) with or without concomitant application of cyclic stretch (10% strain, 0.2 Hz), exhibited a well-polarized and cobblestone-like morphology with increased cell height (27.0 ± 1.3 µm), strongly delineated junctions, and densely packed microvilli. In line with our previous observations, application of the constant flow (shear stress) was critical for promoting maturation of a well-polarized epithelium, while short-term application of cyclic strain did not show any additional effect. Moreover, prolonged cell exposure to flow and cyclic strain resulted in the spontaneous development (around day 6 of culture) of epithelial undulations ('villi-like structures') extending into the lumen of the epithelial channel and covered by continuous brush border (*Figure 1C*). Immunofluorescence confocal analysis confirmed the establishment of confluent epithelial and endothelial monolayers across the entire length of the

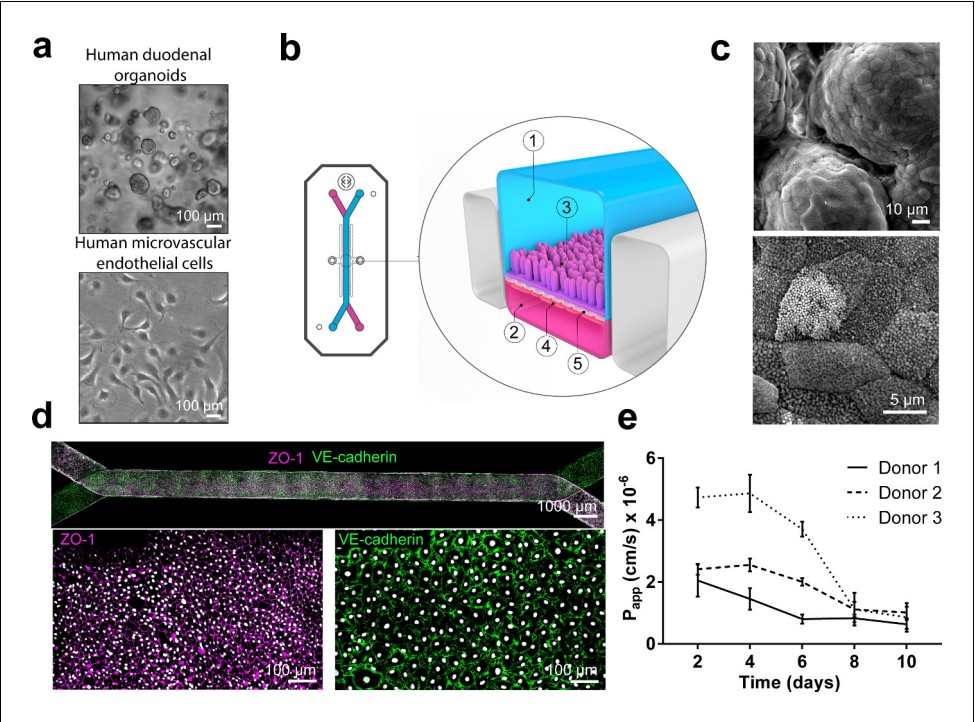

**Figure 1.** Duodenum Intestine-Chip: a microengineered model of the human duodenum. (a) Brightfield images of human duodenal organoids (top) and human microvascular endothelial cells (bottom) acquired before their seeding into epithelial and endothelial channels of the chip, respectively. (b) Schematic representation of Duodenum Intestine-Chip, including its top view (left) and vertical section (right) showing: the epithelial (1; blue) and vascular (2; pink) cell culture microchannels populated by intestinal epithelial cells (3) and endothelial cells (4), respectively, and separated by a flexible, porous, ECM-coated PDMS membrane (5). (c) Scanning electron micrograph showing complex intestinal epithelial tissue architecture achieved by duodenal epithelium grown for 8 days on the chip (top) in the presence of constant flow of media (30 µl/hr) and cyclic membrane deformations (10% strain, 0.2 Hz). High magnification of the apical epithelial cell surface with densely packed intestinal microvilli (bottom). See Figure 1-figure supplement demonstrating the effect of mechanical forces on the cytoarchitecture of epithelial cells and the formation of intestinal microvilli (d) Composite tile scan fluorescence image 8 days post-seeding (top) showing a fully confluent monolayer of organoid-derived intestinal epithelial cells (magenta, ZO-1 staining) lining the lumen of Duodenum Intestine-Chip and interfacing with microvascular endothelium (green, VE-cadherin staining) seeded in the adjacent vascular channel. Higher magnification views of epithelial tight junctions (bottom left) stained against ZO-1 (magenta) and endothelial adherence junctions visualized by VE-cadherin (bottom right) staining. Cells nuclei are shown in gray. Scale bars, 1000 µm (top), 100 µm (bottom) (e) Apparent permeability values of Duodenum Intestine-Chips cultured in the presence of flow and stretch (30 µl/hr; 10% strain, 0.2 Hz) for up to 10 days. $P_{app}$ values were calculated from the diffusion of 3 kDa Dextran from the luminal to the vascular channel. Data represent three independent experiments performed with three different chips/donor, total of three donors; Error bars indicate s.e.m.

The online version of this article includes the following figure supplement(s) for figure 1:

**Figure supplement 1.** Flow-induced increase in primary intestinal epithelial cells height and microvilli formation.

chip (*Figure 1D*), with well-defined epithelial tight junctions, as demonstrated by ZO-1 protein staining and endothelial adherent junctions visualized using antibodies against VE-cadherin (*Dawson et al., 2016*). Importantly, these culture conditions resulted in a time-dependent improvement of intestinal permeability as indicated by the low permeability coefficient ($P_{app}$) of fluorescently labeled dextran recorded in the Duodenum Intestine-Chip generated from organoid-derived cells of three different individuals (*Figure 1E*). Overall, this data indicates that the human adult Duodenum Intestine-Chip supports the formation of a functional barrier with *in vivo* relevant cytoarchitecture, cell-cell interactions, and permeability parameters.

To confirm differentiation of the organoid-derived cells within the chip into all of the distinct epithelial cell lineages as found *in vivo*, we assessed average mRNA gene expression levels of cell-type-

specific markers in the Duodenum Intestine-Chip established from the cells of three different donors. Specific genes included: alkaline phosphatase (*ALPI*) for absorptive enterocytes, mucin 2 (*MUC2*) for goblet cells, chromogranin A (*CHGA*) for enteroendocrine cells, and lysozyme (*LYZ*) for Paneth cells. In addition, we compared the expression levels of these genes in a chip and in freshly isolated adult duodenal tissue (Duodenum). As shown in *Figure 2A*, expression of most of the markers tested, with the exception of lysozyme, increased over time in culture. Notably, on day 8 of chip culture the mRNA expression of alkaline phosphatase and mucin two reached levels similar to

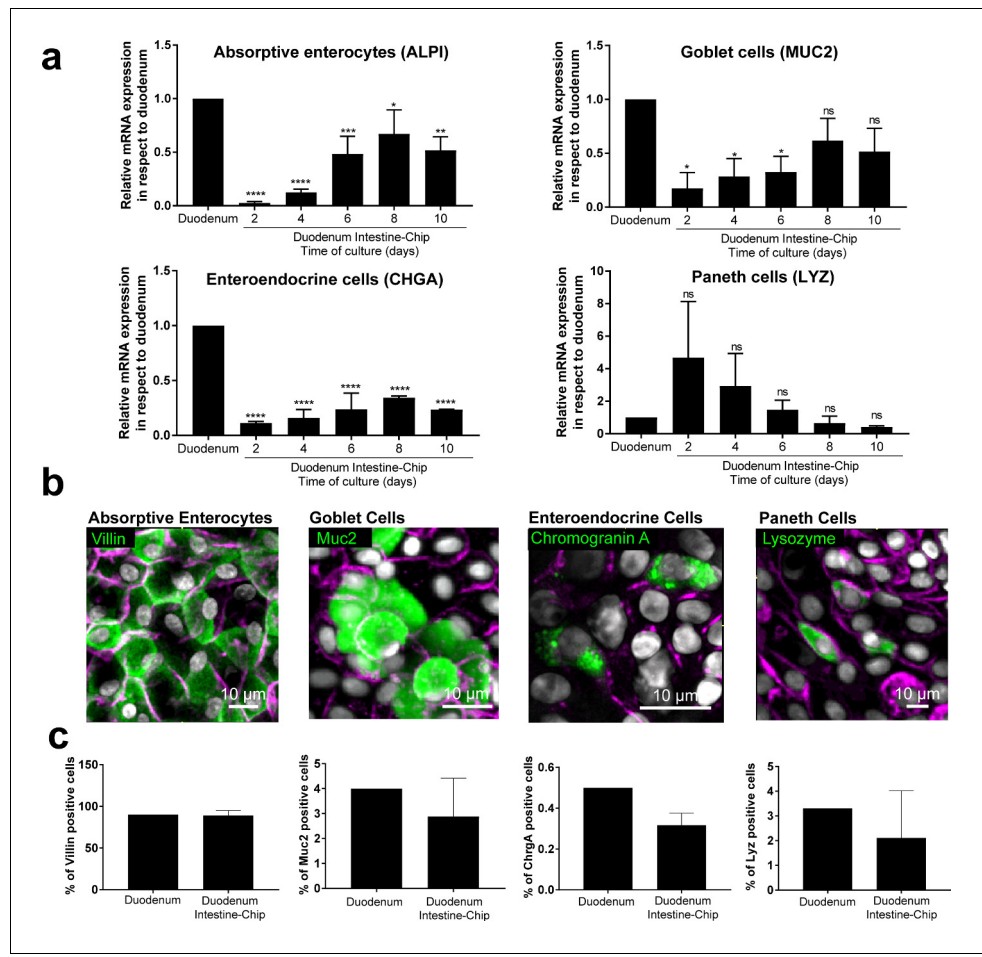

**Figure 2.** Duodenum Intestine-Chip emulates multi-lineage differentiation of native human intestine. (**a**) Comparison of the relative gene expression levels of markers specific for differentiated intestinal cell types, including mucin 2 (*MUC2*) for goblet cells, alkaline phosphatase (*ALPI*) for absorptive enterocytes, chromogranin A (*CHGA*) for enteroendocrine cells, lysozyme (*LYZ*) for Paneth cells in Duodenum Intestine-Chip and RNA isolated directly from the duodenal tissue (Duodenum). Expression of these genes at different time points (days 2, 4, 6, 8, 10) of Duodenum Intestine-Chip culture is shown. In each graph, values represent average gene expression ± s.e. m (error bars) from three independent experiments, each using different donors of biopsy-derived organoids and at least three different chips per time point. Values are shown relative to duodenal tissue expressed as 1. EPCAM expression is used as normalizing control. One-way ANOVA, ****$p < 0.0001$, ***$p < 0.001$, **$p < 0.01$, *$p < 0.05$, ns $p > 0.05$. (**b**) Representative confocal fluorescent micrographs demonstrating the presence of all major intestinal cell types (green) in Duodenum Intestine-Chip at day 8 of fluidic culture, including goblet cells stained with anti-mucin-2; enteroendocrine cells visualized with anti-chromogranin A, absorptive enterocytes stained with anti-villin and Paneth cells labeled with anti-lysozyme. Cell-cell borders were stained with anti-E-cadherin and are shown in magenta. Cells nuclei are shown in gray. Scale bar, 10 μm. (**c**) Quantification of the different intestinal epithelial cell types present in Duodenum Intestine-Chip at day eight and identified by immunostaining, as described in (**b**). Cell ratios are based on 10 different fields of view (10 FOV) counted in three individual chips (each from a different donor) per staining. DAPI staining was used to evaluate the total cell number. Duodenum values, represent cell ratios observed in the histological sections and are based on the literature (*Karam, 1999*).

those detected in RNA isolated directly from the human duodenal tissue. Lysozyme expression showed the opposite trend (declined expression over time in culture) suggesting a reduction in Paneth cell population, which is in line with the observed increased differentiation of the villus epithelium and decreased stemness. In addition, immunostaining followed by quantitative image analysis confirmed the presence of all major differentiated intestinal epithelial cell types and revealed that the relative abundance of these cell types within the chips is similar to their ratios observed in *in vivo* tissue (*Figure 2B*). Indeed, the ratios of absorptive enterocytes, goblet cells, enteroendocrine and Paneth cells observed in the chips on day 8 of fluidic culture, were close to those reported following histopathological evaluation of sections from normal human duodenum (*Figure 2C*) (*Karam, 1999*). Taken together, these results demonstrate the successful establishment of the adult Duodenum Intestine-Chip that closely recreates the barrier function and multilineage differentiation of human intestinal tissue.

## Transcriptomic comparison of the Duodenum Intestine-Chip versus organoids

To further verify whether the Duodenum Intestine-Chip faithfully recapitulates human adult duodenal tissue and to better understand how much it differs from the organoids used for its establishment, we performed RNA-seq analysis (*Figure 3*). We compared global RNA expression data obtained from: i) duodenal organoids (Organoids; n = 3) cultured for 8 days in a conventional plastic-adherent Matrigel drop overlaid with growth medium; ii) Duodenum Intestine-Chip established using cells derived from the above organoids (Duodenum Intestine-Chip; n = 3) and grown for 8 days in the presence of constant flow and stretch; iii) human adult duodenal tissue (Adult Duodenum; n = 2; full-thickness samples) (*Supplementary file 1*). Importantly, the same experimental conditions that is maintenance in expansion media for 6 days, followed by 2 days in differentiation media were used for both organoids and chips. We annotated 13,735 genes in the genome and performed differential gene expression analysis (DGE). For the DGE analysis, we used the 'limma' R package which has an excellent performance even at low sample numbers (*Ritchie et al., 2015*; *Baccarella et al., 2018*). To select the differentially expressed genes, we applied the widely accepted thresholds adjusted p-value<0.05 and |log2FoldChange| > 2. These strict thresholds allowed for a significant increase in the statistical power of the results.

First, we examined the differential gene expression in organoids as compared to human adult duodenal tissue. Out of the 13,735 genes annotated in the genome, 1437 were found to be significantly differentially regulated between these samples: 562 and 875 genes were respectively up- and down-regulated (*Figure 3—figure supplement 1A* and *Figure 3—figure supplement 1—source data 1*). Next, functional enrichment analysis was performed utilizing the PANTHER classification system to highlight biological processes, that is significantly enriched gene ontology (GO) terms within these gene sets (*Ashburner et al., 2000*; *Mi et al., 2013*; *The Gene Ontology Consortium, 2017*; *The Gene Ontology Consortium, 2017*). The majority of differentially expressed genes belonged to pathways related to digestion, extracellular matrix organization, angiogenesis, cell adhesion, tissue development, and cell responses to drugs and xenobiotics (*Figure 3—figure supplement 1B*). This comparison allowed us to identify genes responsible for global transcriptomic differences between organoid technology and native human tissue and highlight biological functions which could be affected by the observed differences.

We applied a similar type of analysis, to compare the global gene expression profiles of the Duodenum Intestine-Chip and adult duodenal tissue. This analysis resulted in the identification of 1023 differentially expressed genes, either significantly up- (382 genes) or down-regulated (641 genes). The significantly smaller number of differentially expressed genes identified with this analysis, as compared to the DGE performed between organoids and adult duodenal tissue, suggests that combining organoids with Organs-on-Chips provides significant advantages to the emulation of the human duodenal tissue (*Figure 3—figure supplement 1B* and *Figure 3—figure supplement 1—source data 2*). Functional enrichment analysis highlighted biological processes, such as protein synthesis and targeting, cell cycle and cell proliferation, underlying the above differences in gene expression profiles.

Next, in order to determine which genes are responsible for the closer similarity of the Duodenum Intestine-Chip, compared to organoids, to human duodenal tissue, we carried out additional DGE analysis. This time, we assessed the differences between Duodenum Intestine-Chip and

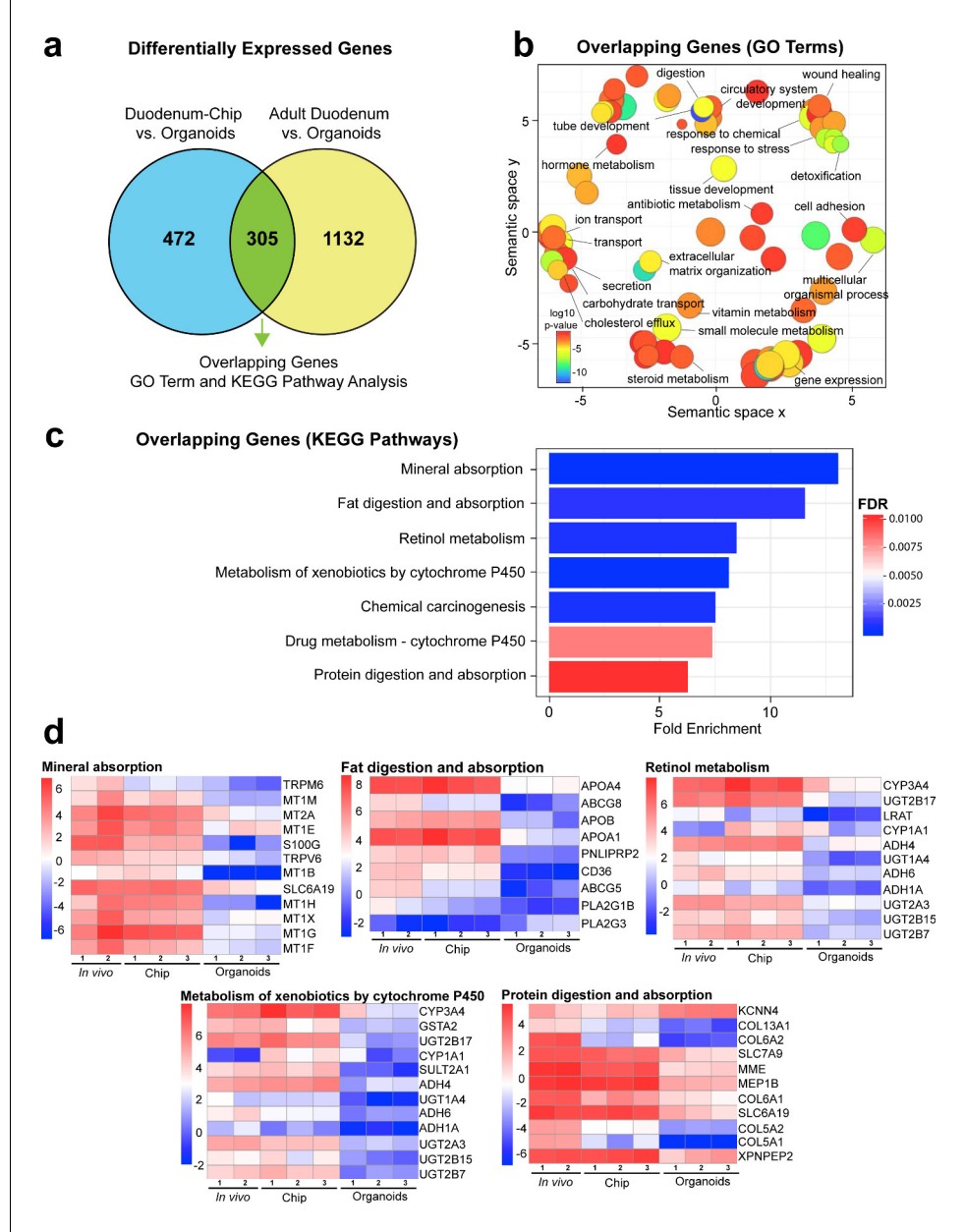

**Figure 3.** Duodenum Intestine-Chip exhibits higher transcriptomic similarity to adult duodenal tissue than organoid culture. (a) Differential gene expression analysis was carried out to identify genes that are up- or down-regulated in Duodenum Intestine-Chip compared to organoids (blue circle) (*Figure 3—source data 1*) and adult duodenum compared to organoids (yellow circle) (*Figure 3—source data 2*). The gene lists were then compared to determine how many genes overlap between those two comparisons (*Figure 3—source data 3*), and the results are shown as a Venn diagram. 305 genes were identified as common and responsible for the closer resemblance of Duodenum Intestine-Chip to human adult duodenum than organoids from which chips were derived. Sample sizes were as follows: Duodenum Intestine-Chip, n = 3 (independent donors); Organoids, n = 3 (independent donors); Adult duodenum, n = 2 (independent biological specimens). Intestinal crypts derived from the same three independent donors were used for the establishment of Duodenum Intestine-Chip and organoid cultures. Both chips and organoids were cultured in parallel, in the presence of expansion media for 6 days, followed by 2 days of differentiation media. Experiment was terminated and samples were processed for analyses 8 days post-seeding. (b) The list of overlapping genes was subjected to GO analysis to identify enriched biological processes (GO terms) (*Figure 3—source data 4*). The results are shown as REVIGO scatterplots in which similar GO terms are grouped in arbitrary two-dimensional space based on semantic similarity. Each circle corresponds to a specific GO term and circle sizes are proportional to the number of genes included in each of the enriched GO

*Figure 3 continued on next page*

*Figure 3 continued*

terms. Finally, the color of a circle indicates the significance of the specific GO term enrichment. GO terms enriched in the overlapping gene set demonstrate that Duodenum Intestine-Chip is more similar to human duodenum with respect to important biological functions of the intestine, including digestion, transport and metabolism. (c) The results of the KEGG pathway analysis using the 305 differentially expressed genes showed seven significantly enriched (FDR adjusted p-value<0.05) pathways related to absorption, metabolism, digestion and chemical carcinogenesis. The size of the bars indicates the fold-enrichment of the corresponding pathways. (d) Curated heatmaps were generated to examine particular genes that belong to the enriched KEGG pathways and to show the expression levels (log$_e$(FPKM)) of these genes across different samples. Genes belonging to five different pathways, including: 'mineral absorption', 'fat digestion and absorption', 'retinol metabolism', 'metabolism of xenobiotics by cytochrome P450' and 'protein digestion and absorption', are shown. The expression levels of genes associated with 'chemical carcinogenesis' (*CYP3A4, GSTA2, UGT2B17, CYP1A1, SULT2A1, ADH4, UGT1A4, ADH6, ADH1A, UGT2A3, UGT2B15, UGT2B7*) and 'drug metabolism – cytochrome 450' *CYP3A4, GSTA2, UGT2B17, ADH4, UGT1A4, ADH6, ADH1A, UGT2A3, UGT2B15,* UGT2B7) were included in the heatmap representing 'metabolism of xenobiotics by cytochrome P450' as they showed to overlap in between three different pathways. Each heatmap has its own color scale, which corresponds to a different range of log$_e$(FPKM) values, as indicated on the color bars located to the left. The provided results further demonstrate that Duodenum Intestine-Chip (Chip) is more similar to adult duodenum (*In vivo*) than are the organoids (Organoids). Sample sizes were as follows: Duodenum Intestine-Chip, n = 3 (independent donors); Organoids, n = 3 (independent donors); *In vivo*, adult duodenum, n = 2 (independent biological specimens). Intestinal crypts derived from the same three independent donors were used for the establishment of Duodenum Intestine-Chip and organoid cultures. Both chips and organoids were cultured in parallel as described in (a). See also *Figure 3— figure supplement 1*, *Figure 3—figure supplement 1—source data 1* and *Figure 3—figure supplement 1— source data 2* showing the results of DGE analysis followed by functional enrichment performed between Organoids or Duodenum Intestine-Chip and Adult Duodenum.

The online version of this article includes the following source data and figure supplement(s) for figure 3:

**Source data 1.** Differentially expressed genes in Duodenum Intestine-Chip vs. duodenal organoids, related to *Figure 3*.

**Source data 2.** Differentially expressed genes in adult duodenum vs. duodenal organoids, related to *Figure 3*.

**Source data 3.** Differentially expressed genes common in Duodenum Intestine-Chip and adult duodenum versus duodenal organoids, related to *Figure 3*.

**Source data 4.** Enriched GO terms from a list of differentially expressed genes common in Duodenum Intestine-Chip and adult intestine versus duodenal organoids, related to *Figure 3*.

**Figure supplement 1.** Differentially expressed genes and enriched gene ontology categories in organoids or Duodenum Intestine-Chip with respect to adult duodenum.

**Figure supplement 1—source data 1.** Differentially expressed genes in duodenal organoids vs. adult duodenum, related to *Figure 3—figure supplement 1*.

**Figure supplement 1—source data 2.** Differentially expressed genes in Duodenum Intestine-Chip vs. adult duodenum, related to *Figure 3—figure supplement 1*.

organoids, and examined these differences relative to those that exist when comparing the adult human duodenal tissue and organoids (Duodenum Intestine-Chip versus Organoids and human Adult duodenum versus Organoids). We identified genes that were significantly up- and down-regulated in the Duodenum Intestine-Chip relative to the organoid culture (*Figure 3—source data 1*) and identified the proportion of these genes that were also significantly different in adult duodenal tissue relative to organoids (*Figure 3—source data 2*). We found 305 genes which were common for Duodenum Intestine-Chip and adult duodenum but different from organoids, representing 39.25% of differentially expressed genes in Duodenum Intestine-Chip versus organoids comparison, and 21.22% of differentially expressed genes in human adult duodenal tissue versus organoids comparison (*Figure 3A* and *Figure 3—source data 3*). To further explore overlapping genes, we used the Gene Ontology and Kyoto Encyclopedia of Genes and Genomes (KEGG) Pathway Analysis to examine biological processes and pathways that are enriched in this gene list (*Kanehisa and Goto, 2000*; *Thomas et al., 2003*). In addition, we used the well-established Reduce and Visualize Gene Ontology (REVIGO) tool to reduce redundancies by clustering semantically related GO terms and outputting them as a scatter plot for visualization (*Eden et al., 2007*; *Supek et al., 2011*). As a result, the list of 305 overlapping genes was associated with 117 significant GO terms, which were reduced to 74 unique GO terms in REVIGO (*Figure 3—source data 4*). Notably, biological

processes enriched in this gene set were associated with important intestinal functions such as digestion and transport of nutrients and ions, metabolism, detoxification, as well as tissue and tube development (*Figure 3B*).

Additionally, KEGG Pathway analysis identified a total of seven significantly enriched pathways (threshold FDR adjusted p-value<0.05) among the overlapping genes (*Figure 3C*). The corresponding fold enrichments of these pathways are shown in *Figure 3C*. For the identification of the enriched pathways, we used the FDR adjusted p-value (instead of the p-value) which significantly increased the statistical power of our findings. Importantly, many of these pathways were linked to drug metabolism, such as – cytochrome CYP450 (associated with 10 genes), metabolism of xenobiotic by cytochrome CYP450 (associated with 12 genes), and other important pathways related to nutrients absorption and digestion, such as mineral absorption (associated with 12 genes), fat digestion and absorption (associated with nine genes), protein digestion and absorption (associated with 11 genes) (*Figure 3C*). To more closely examine gene signatures that belong to these pathways, we used $\log_e$(FPKM) values generated from the RNA-seq dataset, and generated heatmaps for differentially expressed genes that fall under the seven significantly enriched pathways (*Figure 3D*). A large number of genes involved in the nutrient absorption, digestion, as well as, metabolism of xenobiotics were highly expressed in Duodenum Intestine-Chip (Chip) and adult duodenum tissue (*In vivo*), but expressed at the low level in organoids (Organoids).

Cumulatively, this data demonstrated an increased similarity in the global gene expression profile between Duodenum Intestine-Chip and human adult duodenal tissue, compared to organoids.

## Intestinal drug transporters and MDR1 efflux activity in the Duodenum Intestine-Chip

Encouraged by the results of transcriptomic analyses detailed above we sought to focus further on the characterization of the expression, localization and function of the main intestinal drug transporters. First, we assessed the expression and localization of these transporters in Duodenum Intestine-Chip established from the organoid-derived cells of three independent donors by qRT-PCR and immunofluorescent imaging. The average gene expression levels of efflux (MDR1, BCRP, MRP2, MRP3) and uptake (PepT1, OATP2B1, OCT1, SLC40A1) drug transporters, were assessed at day 8 of chip culture and compared with their levels observed in the freshly isolated human duodenal tissue (Duodenum) and the previously described Intestine-Chip model based on the use of Caco-2 cells (Caco-2 Intestine-Chip) (*Figure 4A*). The expression profiles of these genes in Duodenum Intestine-Chip were similar to those observed in human duodenal tissue. Although a similar pattern of gene expression was also found in the Caco-2 Intestine-Chip (day 8 of culture), expression of a couple of important organic anion and cation transporters, including OATP2B1 and OCT1, were markedly increased in comparison to the human duodenum. This data confirms that the well-known differences between Caco-2 cells and normal human intestinal tissue are maintained also in the case of on chip culture (*Maubon et al., 2007*; *Ölander et al., 2016*). Moreover, although not statistically significant, notable differences in the expression of MDR1, BCRP and PEPT1 were found in Caco-2 Intestine-Chip in comparison to Duodenum. In line with the trends previously reported by others (*Sun et al., 2002*; *Maubon et al., 2007*; *Harwood et al., 2016*), a 3.5-fold higher expression of MDR1, 3.6-fold and 24-fold lower expression of BCRP and PEPT1, respectively, were found in Caco-2 cells compared with the human duodenum. On the other hand, much smaller differences in respect to native human tissue were observed for Duodenum Intestine-Chip: with ~1.5 fold and ~2.6 fold increased expression of MDR1 and BCRP, respectively, and no differences noted in the expression of PEPT1. These results demonstrate that by combining duodenal organoids with Organs-on-Chips technology we enabled a closer emulation of human *in vivo* tissue gene expression profile, than with the previously described Caco-2 Intestine-Chip.

We further demonstrated *in vivo* relevant localization of the luminal efflux pumps MDR1, more commonly referred to as P-gp or P-glycoprotein (*Figure 4B*) and BCRP (*Figure 4—figure supplement 1A*) as well as the uptake Peptide Transporter 1 (PEPT1) in Duodenum Intestine-Chip (*Figure 4—figure supplement 1C*). All three transporters showed to co-distribute together with villin, a marker specific for apical cell membrane, to the intestinal cell brush border of Duodenum Intestine-Chip. This was confirmed on the cross-sectional confocal images of the duodenal epithelium cultured on chip showing a co-localization (merge channel; white) of the fluorescent signal of MDR1, BCRP or PEPT1 with that of villin staining. Similarly, line plots depicting the distribution of fluorescence

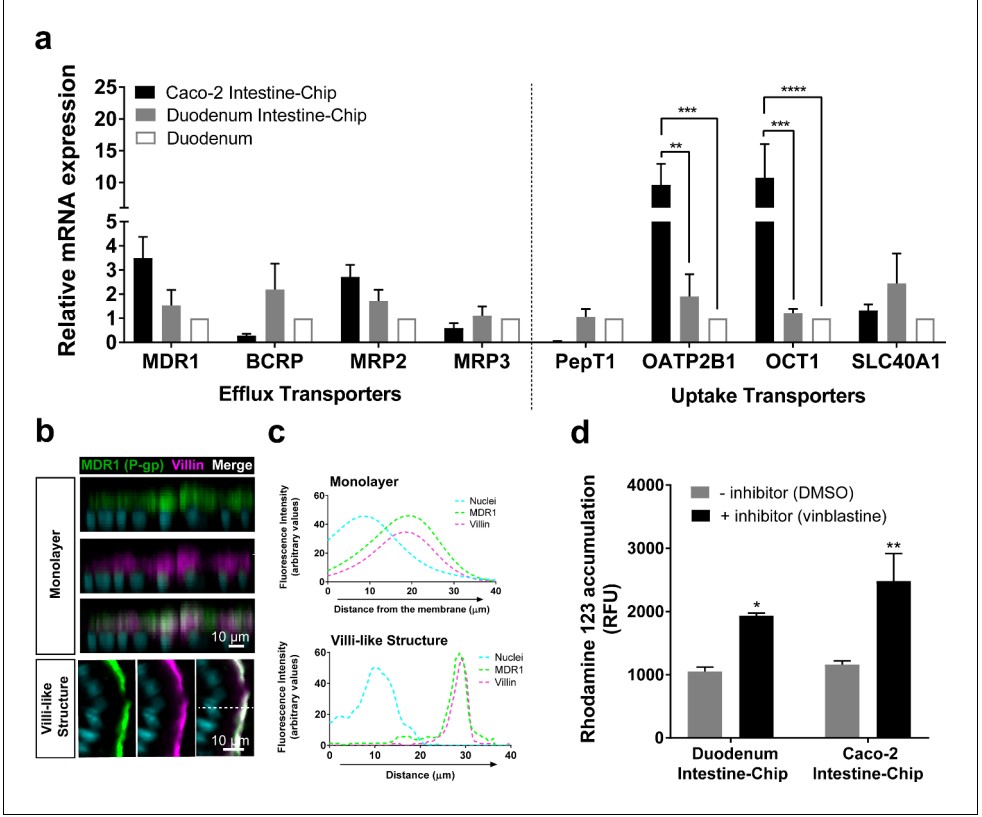

**Figure 4.** Duodenum Intestine-Chip shows the presence of major intestinal drug transporters and correct localization and function of efflux pump MDR1 (P–gp). (**a**) Comparison of the relative average gene expression levels of drug efflux (MDR1, BCRP, MRP2, MRP3) and uptake (PEPT1, OATP2B1, OCT1, SLC40A1) transporters in Caco-2 Intestine-Chip, Duodenum Intestine-Chip, both assessed on day 8 of culture, and RNA isolated directly from the duodenal tissue (Duodenum). The results show that Duodenum Intestine-Chip expresses drug transport proteins at the levels close to human duodenal tissue. Note, the expression of OATP2B1 and OCT1 in Caco-2 Intestine-Chip were significantly higher than in human duodenum while the difference between Duodenum Intestine-Chip and adult duodenum is not significant. Each value represents average gene expression ± s.e.m (error bars) from three independent experiments, each involving Duodenum Intestine-Chip established from a tissue of three different donors (three chips/donor), RNA tissue from three independent biological specimens, and Caco-2 Intestine-Chip (three chips). Values are shown relative to the duodenal tissue expressed as 1, two-way ANOVA, ****p<0.0001, ***p<0.001, **p<0.01. EPCAM expression was used as normalizing control. (**b**) Representative confocal immunofluorescence micrographs showing apical localization of the efflux transporter MDR1 (green) and the cell surface marker villin (magenta) in a vertical cross section of monolayer (top) formed in Duodenum Intestine-Chip at day 4 and later formed villi-like structure (bottom) at day 8. Cell nuclei are visualized in cyan. Scale bar, 10 μm. (**c**) Line plots corresponding to confocal images in (**b**) showing the distribution of fluorescent intensities for three different channels: MDR1 (green), villin (magenta) and nuclei (cyan) along the basal–apical axis of enterocytes forming a monolayer or villi-like structures in Duodenum Intestine-Chip. The fluorescent intensity was analyzed in 3D reconstructed confocal images of Duodenum Intestine-Chip and plotted as average across 20 different z-stacks. Distribution of MDR1 and villin shows significant overlap. See also *Figure 4—figure supplement 1* showing luminal localization of additional efflux (BCRP) and uptake (PEPT1) transporters in Duodenum Intestine-Chip. (**d**) Activity of efflux pump proteins in Caco-2 and Duodenum Intestine-Chip. The intracellular accumulation of the fluorescent substrate of MDR1 - Rhodamine 123 is significantly increased in response to the MDR1 inhibitor vinblastine (black bars) in comparison to vehicle (DMSO) control (gray bars) in Caco2 and Duodenum Intestine-Chip. Data were represented as mean ± s.e.m (error bars) of at least three independent experiments involving chips generated from organoids of three individual donors or Caco-2 cells, all assessed 8 days post-seeding. Two-way ANOVA, **p<0.01, *p<0.05.

The online version of this article includes the following figure supplement(s) for figure 4:

**Figure supplement 1.** Luminal localization of efflux (BCRP) and uptake (PEPT1) transporters in Duodenum Intestine-Chip.

intensities of each transporter and villin along the basal-apical cell axis revealed the presence of significant overlap between the two channels (*Figure 4C* and *Figure 4—figure supplement 1*). Notably, the physiologically relevant localization of MDR1, BCRP1 and PEPT1 at the luminal surface of intestinal epithelium was confirmed at two different time-points of Duodenum Intestine-Chip culture – when the cells have formed a confluent monolayer (around day 4 of culture) and after the villi-like structures morphogenesis has occurred (at day 8 of culture). The MDR1 activity was confirmed by measuring the intracellular accumulation of rhodamine 123 in the presence and absence of specific MDR1 inhibitor, vinblastine, across Duodenum Intestine-Chip at day 8 of their culture. In addition, the obtained results were compared to MDR1 activity assessed in the Caco-2 Intestine-Chip model (*Figure 4D*). Exposure to the inhibitor induced a ~ 2 fold increase in intracellular accumulation of rhodamine (1.84-fold increase in Duodenum Intestine-Chip and 2.14-fold increase in Caco-2 cells-based model), confirming the presence of active MDR1 efflux pumps in both cell systems.

## Drug-mediated CYP3A4 induction in the Duodenum Intestine-Chip

Induction of CYP450 drug metabolizing enzymes in human intestine is a major concern for the pharmaceutical industry, as it is known to impact the pharmacokinetics and bioavailability of various orally administered drugs, as well as, mediate DDIs. Therefore, we evaluated the capability of our Duodenum Intestine-Chip to be applied for CYP3A4 induction studies and to help identify risk for DDIs in the clinic. This is not feasible in pre-clinical species, such as rat and dog, due to marked species differences in the expression and regulation of cytochrome P450, as well as substrate specificity of the nuclear receptors, such as PXR, responsible for the transcriptional regulation of CYP3A4 and several drug transporters. Caco-2 Intestine-Chip has been previously reported to possess an increased activity of the CYP450 enzymes when compared to the conventional static culture of Caco-2 cells on transwell inserts (*Kim and Ingber, 2013*). However, the gene expression level of *CYP3A4* measured in this system is significantly lower than in the adult human intestine (*Figure 5A*), limiting its application for pharmaceutical research, specifically pharmacokinetic evaluation. In the present study, we demonstrate that, in comparison to the Caco-2 cells based system, our Duodenum Intestine-Chip expressed CYP3A4 at a much higher gene (~6000 times higher, p<0.0001) (*Figure 5A*) and protein level (*Figure 5B*), similar to those observed in the adult human duodenum.

In order to assess the drug-mediated CYP3A4 induction potential in the Duodenum Intestine-Chip, we exposed it to rifampicin (RIF) and 1,25-dihidroxyvitamin D3 (VD3), both of which are known prototypical CYP3A4 inducers. Treatment of Duodenum Intestine-Chip from all three donors with RIF and VD3 resulted in 5.3-fold and 4.1-fold induction of *CYP3A4* expression, respectively, relative to that in the DMSO-treated controls (*Figure 5C*, left). Alternatively, Caco-2 Intestine-Chip was shown to respond to VD3, but not RIF treatment (*Figure 5C*, right), which is consistent with the previously published reports (*Schmiedlin-Ren et al., 1997*; *Ozawa et al., 2015*; *Negoro et al., 2016*). Dramatically low baseline expression of *CYP3A4* in Caco-2 cells derived system showed to increase by 316-fold in the presence of VD3. However, it did not reach the levels observed in the human tissue and remained undetectable at the protein level as assessed by western blot analysis (*Figure 5C*, below graph). The metabolism of testosterone to 6β-hydroxytestosterone (6β-OH-T), was used to evaluate CYP3A enzyme activity in Caco-2 and Duodenum Intestine-Chip (*Figure 5D*). Consistent with the gene and protein expression data, the formation of 6β-OH-T was higher in Duodenum Intestine-Chip (55.6 ± 7.8 pmol/min/mg protein) compared with Caco-2 Intestine-Chip (4.9 ± 0.64 pmol/min/mg protein), and was further increased by RIF (214.4 ± 106.5 pmol/min/mg protein) and VD3 treatment (198.0 ± 41.9 pmol/min/mg protein). The average increase in the metabolic activity of Duodenum Intestine-Chip was 3.9-fold for RIF and 3.6-fold for VD3 compared with DMSO controls. On the other hand, none of the studied prototypical inducers showed an effect on CYP3A4-mediated hydroxylation of testosterone in Caco-2 Intestine-Chip. The observed differences between drug-mediated induction of CYP3A4 between Duodenum and Caco-2 Intestine-Chip could be attributed to differences in gene expression levels of intestinal nuclear receptors, including PXR and vitamin D receptor (VDR) (*Figure 5E*). The gene expression levels of *PXR* and *VDR* in Duodenum Intestine-Chip were significantly higher and closer to human duodenal tissue than those in Caco-2 Intestine-Chip. The demonstrated lack of *PXR*, known to be one of the key transcriptional regulators of CYP3A4 induction in humans, in Caco-2 cells may explain the differences between the Duodenum Intestine-Chip and other Caco-2 cells–based systems, including Caco-2 Intestine-Chip. Altogether, our findings support the conclusion that the Duodenum Intestine-Chip represents an improved

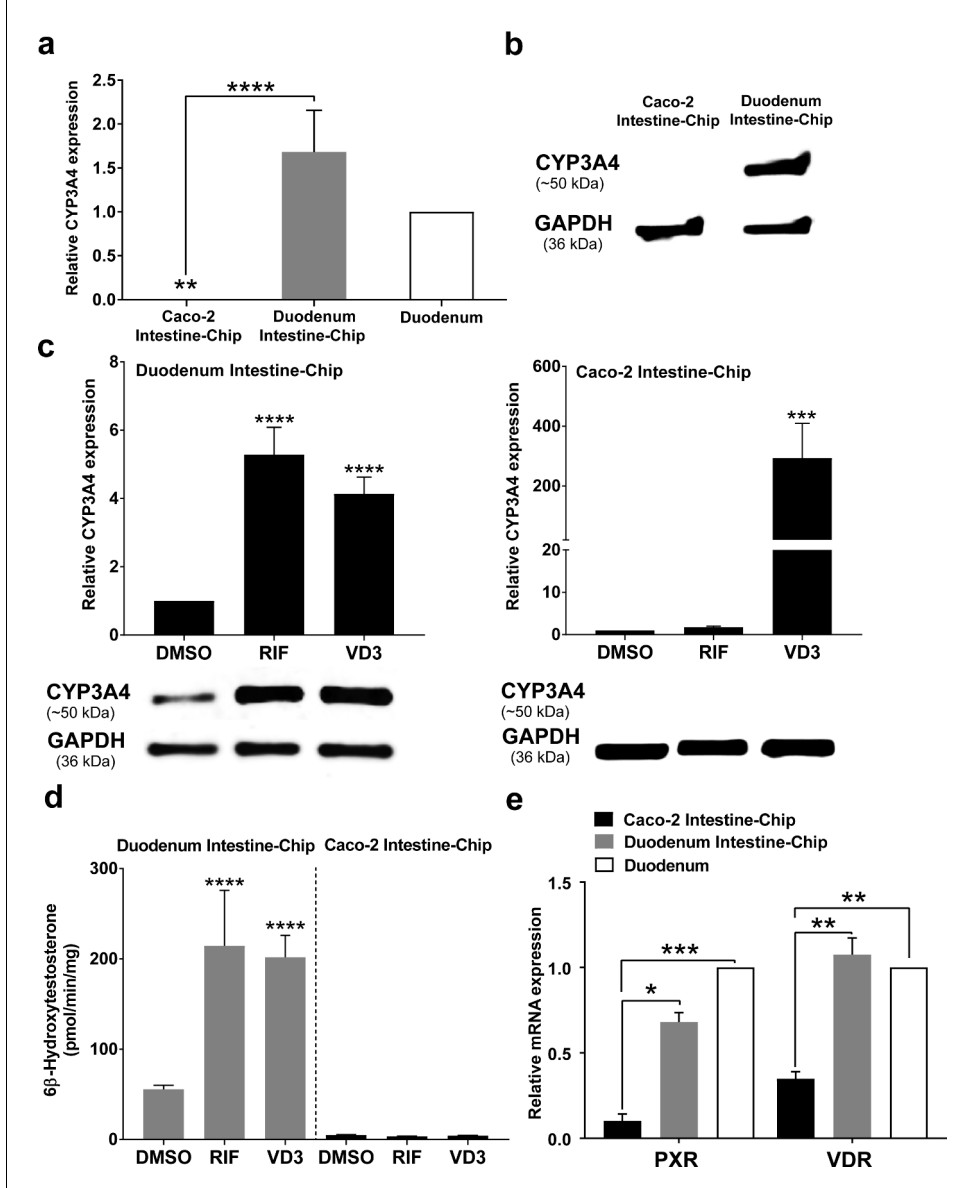

**Figure 5.** CYP3A4 expression levels and induction in Duodenum Intestine-Chip and Caco-2 Intestine-Chip. (a) Average gene expression levels of CYP3A4 ± s.e.m (error bars) in Caco-2 Intestine-Chip, Duodenum Intestine-Chip assessed at day 8 of their culture and human duodenum (three independent biological specimens). All values are shown relative to the adult duodenal tissue expressed as 1, one-way ANOVA, ****p<0.0001, **p<0.01. EPCAM expression was used as normalizing control. (b) Protein analysis of CYP3A4 in Caco-2 and Duodenum Intestine-Chip assessed at day eight using western blot. (c) The CYP3A4 induction in Duodenum Intestine-Chip and Caco-2 Intestine-Chip treated at day 6 with solvent (DMSO), 20 µM rifampicin (RIF) or 100 nM 1,25-dihidroxyvitamin D3 (VD3) for 48 hr. The gene expression levels (top) of CYP3A4 were examined by real-time PCR analysis at day 8. On the y-axis, the gene expression levels in the DMSO-treated chips were taken as 1.0. All data are represented as means ± s.e.m. Two-way ANOVA, ****p<0.0001 (compared with DMSO-treated cells). The corresponding CYP3A4 protein expression levels (bottom) were measured at day 8 by western blotting analysis. (d) CYP3A4 enzyme activity was determined by monitoring the formation of 6β-hydroxytestosterone in the medium of Duodenum Intestine-Chip and Caco-2 Intestine-Chip, as measured by LC-MS. For induction studies, 20 µM RIF or 100 nM VD3 was added 48 hr before measurement. Data are expressed as mean ± s.e.m of three independent experiments each involving Duodenum Intestine-Chip established from organoid-derived cells of a different donor and Caco-2 Intestine-Chip. At least three different chips were used per condition. Two-way ANOVA, ****p<0.0001 (e) Gene expression analysis of the receptors PXR and VDR in Duodenum Intestine-Chip and Caco-2 Intestine-Chip examined at day 8 by real-time PCR analysis and compared to their expression in adult duodenal tissue

*Figure 5 continued on next page*

*Figure 5 continued*

(Duodenum). On the y-axis, the gene expression levels in adult tissue were taken as 1.0. All data are represented as means ± s.e.m. Two-way ANOVA, \*\*\*p<0.001, \*\*p<0.01, \*p<0.05.

human-relevant model for predicting CYP3A4-mediated drug-drug interactions in the intestine for orally delivered drugs, than the commonly used Caco-2 cell-based models.

## Discussion

A number of *in vitro* and animal models have been developed and routinely applied for characterization and prediction of absorption, metabolism, and excretion of xenobiotics in humans. Among these, one of the most widely used models is the conventional Caco-2 monolayer culture, considered as the current 'gold standard' in studying intestinal disposition of drugs *in vitro*. However, inherent limitations, such as lack of *in vivo* relevant 3D cytoarchitecture, lack of appropriate ratio of cell populations, altered expression profiles of drug transporters and drug metabolizing enzymes, especially CYP450s, and aberrant CYP450 induction response, challenge the use of these model for predicting ADME in the clinic. On the other side of the spectrum, animal models, while retaining proper physiological conditions, exhibit species differences in both drug metabolism and drug transport, as well as substrate specificity for nuclear receptors regulating CYP450s and transporters. For these reasons, there is a need for new models for predicting human ADME and determining risk for DDIs mediated by intestinal CYP450s and drug transporters.

Current preclinical models are not able to fully recapitulate the complex nature and function of human intestinal tissue, leading to a limited accuracy and poor predictability in drug development. In this study, we leveraged Organs-on-Chips technology and primary adult duodenal organoids to emulate multicellular complexity, physiological environment, native intestinal tissue architecture and functions and to create an alternative human-relevant model for the assessment of ADME of orally administered drugs. Since clinical pharmacology studies are typically conducted in adult individuals and it is well known that pharmacokinetics in children are very different than in adults with respect to drug absorption, distribution, metabolism, and elimination (*Grimsrud et al., 2015*), we used organoids isolated from adult healthy donors to develop Duodenum Intestine-Chip. Our new findings, presented here, complement the methodology and characterization of the Intestine-Chip model established in our previous report (*Kasendra et al., 2018*) using children biopsies-derived organoids. We demonstrate that the physiological mimicry of the Duodenum Intestine-Chip is excellent and superior to the organoids in their original form (static, not on chip). The synergistic use of organoid-derived cells and Organs-on-Chips enables the establishment of intact tissue-tissue interface formed by adult intestinal epithelium and small intestinal microvascular endothelial cells. The application of mechanical stimulation in the form of continuous luminal and vascular flow, showed to exert beneficial effects on the intestinal tissue architecture. Increased epithelial cell height, acquisition of cobblestone-like cell morphology, formation of well-defined cell-cell junctions, and dense intestinal microvilli were attributed to the presence of flow. We confirmed the development of comparable levels of intestinal barrier function to hydrophilic solute (dextran) across Duodenum Intestine-Chip cultures established from organoid-derived cells of three independent donors. This functional feature of intestinal tissue becomes critical when studying drug uptake, efflux, and disposition in polarized organ systems. Although many studies have demonstrated that Caco-2 transwell allow good prediction of transcellular drug absorption, they showed to be unreliable in the assessment of passive diffusion of polar molecules such as hydrophilic drugs and peptides. This is related to much smaller, in respect to small intestinal villi, effective area of the monolayer and its higher tight junctional resistance (*Press and Di Grandi, 2008*; *Kumar et al., 2010*). Organoids have quickly become a model of interest in drug discovery and development applications, yet the use of organoids for the studies of intestinal permeability has been linked with major technical difficulties related to their 3D structure, limited access to lumen, and the need for the use of sophisticated microinjection techniques for the application of the compound at the apical cell surface. On the other hand, Duodenum Intestine-Chip emulates complex intestinal tissue architecture, as evidenced by the presence of 3D villi-like structures on scanning electron micrographs acquired on day 8 of fluidic culture, while allowing a direct exposure of apical cell surface to test compounds. This represents a major technological

advantage over Caco-2 transwell and organoids and might provide a valid alternative for the preclinical studies of intestinal drug absorption.

Immunofluorescent imaging and gene expression analysis demonstrated that Duodenum Intestine-Chip possess specialized intestinal cell subpopulations that are absent in tumor-derived cell lines including Caco-2. Importantly, they are present in the Duodenum Intestine-Chip at the physiologically relevant ratios, similar to the ones observed in the native human duodenum. Expression of the markers specific for the intestinal types that naturally reside in the villus compartment – alkaline phosphatase for absorptive enterocytes, mucin two for goblet cells, chromogranin A for enteroendocrine cells – showed to increase up to day 8 of Duodenum Intestine-Chip culture. This has been accompanied by the decline in the expression of genes specific for crypt-residing cells – lysozyme for Paneth cells and Lgr5 for stem cells (data not shown) suggesting the acquisition of terminally differentiated cell phenotype by all of the cells present in the chip. This is in line with well-known effect of Wnt3a withdrawal on organoids-derived primary intestinal epithelium (*Sato et al., 2009*; *Sato et al., 2011*). Because human intestinal cells do not produce significant amounts of Wnt3a, EGF, or several other growth factors essential for stem cell division and cell proliferation, removal of Wnt3a supplementation leads to a loss of Lgr5-positive cells, decreased cell proliferation, appearance of secretory cell lineages, including goblet and enteroendocrine cells, and the transformation of immature crypt-like enterocytes into differentiated nutrient-absorptive cells. The presence of differentiated cell subpopulations is critical for modeling various aspects of intestinal biology and function, including mucus production, antimicrobial response, host-microbiome interaction, secretion of intestinal hormones, nutrients digestion and absorption. Given that we have shown that the Duodenum Intestine-Chip recapitulates faithfully the physiologically relevant ratios of these cells it could be readily applied beyond ADME applications in development of new approaches and therapeutics that target specific cell populations. For example, it could serve to evaluate specific targeting of Paneth cells, which through a series of genetic studies have been implicated in inflammatory bowel disease (*Xavier and Podolsky, 2007*; *Khor et al., 2011*; *Adolph et al., 2013*; *Liu et al., 2016b*).

We compared global RNA expression profiles, obtained by RNA-seq, of Duodenum Intestine-Chip, duodenal organoids and human native duodenal tissue in order to determine which of the two models better reflects their natural counterpart. Surprisingly, although chips and organoids were established with the cells of the same origin (donors, source of the tissue) their transcriptomic profiles were shown to significantly differ from each other – the total number of 472 up- and down-regulated genes were found in this comparison. Moreover, several different analyses demonstrated that the transcriptome of Duodenum Intestine-Chip more closely resembles global gene expression in human adult duodenum than do the organoids, strongly suggesting that Duodenum Intestine-Chip constitute a closer approximation of human *in vivo* tissue. Importantly, the subset of 305 genes were found to be common for Duodenum Intestine-Chip and human tissue but different from organoids, highlighting the beneficial changes in the transcriptional profile of duodenal organoids achieved by combining them with Organs-on-Chips. These genes showed to be associated with important biological functions including: digestion, transport of nutrients and ions, extracellular matrix organization, wound healing, metabolism, detoxification, and tissue development. In addition, several genes involved in drug metabolism, including but not limited to drug metabolism enzymes: *CYP3A4, UGT1A4, UGT2A3, UGT2B7, UGT2B15, UGT2B17*, were observed to exhibit similar pattern of expression in Duodenum Intestine-Chip and adult duodenal tissue while they differed from organoids, suggesting the improved potential of this system to study biotransformation of xenobiotics and DDIs.

Intestinal efflux and uptake transporters are key determinants of absorption and subsequent bioavailability of a large number of orally administered drugs. *In vivo*-like expression of the major intestinal drug transporters, including clinically relevant *MDR1, BCRP* and *PEPT1*, was demonstrated in Duodenum Intestine-Chip system. Additionally, we confirmed their apical localization, which is known to be crucial for the unique gatekeeper function of these proteins in controlling drug access to metabolizing enzymes and excretory pathways (*Shugarts and Benet, 2009*; *Estudante et al., 2013*), within the plasma membrane of intestinal epithelial cells grown on chip. Functional assessment of MDR1 efflux revealed similar level of activity as observed in Caco-2 Intestine-Chip and its successful inhibition by vinblastine. Notably, in comparison to Caco-2 Intestine-Chip the chip system established using duodenal organoid-derived cells showed improved relative mRNA expression levels of organic anion and cation transporters: *OATP2B1* and *OCT1*. These transporters are

responsible for the uptake of numerous xenobiotics, including statins, antivirals, antibiotics, and anti-cancer drugs (*Roth et al., 2012*). While significantly higher expression of these proteins, in comparison to human duodenal tissue, was observed in Caco-2 model, their levels showed to be closer to *in vivo* in Duodenum Intestine-Chip. Our findings suggest that Duodenum Intestine-Chip system could be applied to assess the specific contribution of efflux transporters to drug disposition, to evaluate the active transport of xenobiotics across intestinal barrier, as well as, modeling increased absorption by targeting of specific uptake transporters such as PEPT1 or OATP2B1.

Evaluation of the expression, activity and drug-mediated induction of the CYP3A4, a major enzyme involved in human metabolism of xenobiotics, revealed a key advantage of Duodenum Intestine-Chip over Caco-2 models. Significantly higher and much closer to *in vivo* expression of CYP3A4 along with increased metabolism of testosterone (6β-hydroxylation) were observed in Duodenum Intestine-Chip in comparison to Caco-2 cells-based system. Consistent with the results of others (*Negoro et al., 2016*) CYP3A4 was undetectable at the protein level in Caco-2 cells and remained unchanged upon stimulation with RIF (a PXR agonist). Additionally, while vitamin D3 stimulation led to a significant increase in the mRNA expression of cytochrome P450 in Caco-2 Intestine-Chip, it showed no effect on the metabolic activity of this enzyme. In contrast, successful and reproducible modulation of the expression and enzymatic activity of CYP3A4 using agonists specific for PXR and VDR, rifampicin and vitamin D3, respectively, was achieved in Duodenum Intestine-Chip engineered individually from the organoid-derived cells of three independent donors. CYP3A4 induction levels observed were similar to those shown for intestinal slices (*van de Kerkhof et al., 2008*), suggesting suitability of this model for the studies of drug metabolism that cannot be supported by the previous Caco-2 models.

In conclusion, Duodenum Intestine-Chip provides a closer to *in vivo* model of the human duodenum, compared to organoids or Caco-2 cells, and represents a potential tool for preclinical drug assessment in a more human-relevant model. Moreover, as it is composed of cells isolated from individual patients, it could be personalized as needed, in order to assess interindividual differences in drug disposition and responses, study the effect of genetic polymorphisms on pharmacokinetics and pharmacodynamics, as well as decoupling the effect of various factors such as age, sex, disease state, and diet on metabolism, clearance, and bioavailability of xenobiotics. This system could also help us to better understand the basic biology of human intestinal tissue in healthy and disease states and potentially enable novel therapeutic development as we further our understanding of the mechanisms driving key disease phenotypes.

# Materials and methods

## Key resources table

| Reagent type (species) or resource | Designation | Source or reference | Identifiers | Additional information |
|---|---|---|---|---|
| Antibody | anti-BCRP (Mouse monoclonal) | Millipore | Cat# MAB4155 RRID:AB_95060 | IF(1:100) |
| Antibody | anti-Chromogranin A (Goat polyclonal) | Santa Cruz Biotechnology | Cat# sc-1488 RRID:AB_2276319 | IF(1:100) |
| Antibody | anti-E-Cadherin (Mouse monoclonal) | Abcam | Cat#: ab1416 RRID:AB_300946 | IF(1:100) |
| Antibody | anti-Lysozyme (Rabbit polyclonal) | Agilent | Cat#: A0099 RRID:AB_2341230 | IF(1:500) |
| Antibody | anti-MDR-1 (P-gp) (Mouse monoclonal) | Thermo Fisher Scientific | Cat# MA5-13854 RRID:AB_10979045 | IF(1:100) |
| Antibody | anti-Mucin-2 (Mouse monoclonal) | Santa Cruz Biotechnology | Cat# sc-7314 RRID:AB_627970 | IF(1:400) |
| Antibody | anti-PEPT1 (Mouse monoclonal) | Santa Cruz Biotechnology | Cat# sc-373742 RRID:AB_10918256 | IF(1:100) |
| Antibody | anti-VE-Cadherin (Rabbit polyclonal) | Abcam | Cat#: ab33168 RRID:AB_870662 | IF(1:400) |

*Continued on next page*

*Continued*

| Reagent type (species) or resource | Designation | Source or reference | Identifiers | Additional information |
|---|---|---|---|---|
| Antibody | anti-Villin (Rabbit monoclonal) | Abcam | Cat#: ab130751 AB_11159755 | IF(1:100) |
| Antibody | anti-ZO-1) (Mouse monoclonal) | Thermo Fisher Scientific | Cat# 33–9100 RRID:AB_2533147 | IF(1:100) |
| Chemical compound, drug | 3 kDa Dextran, Cascade Blue | Thermo Fisher Scientific | Cat# D7132 | 0.1 mg/ml |
| Chemical compound, drug | 1α,25-Dihydroxyvitamin D3 | Sigma | Cat# D1530 | 100 nM |
| Chemical compound, drug | Rifampicin | Sigma | Cat# R3501 | 20 µM |
| Chemical compound, drug | Testosterone | Sigma | Cat# T1500 | 200 µM |
| Commercial assay or kit | MDR1 Efflux Assay Kit | Millipore | Cat# ECM910 | |
| Software, algorithm | Prism | GraphPad | | |
| Software, algorithm | Fiji | | RRID:SCR_002285 | |

## Human tissue collection, generation, and culture of organoids

Human duodenal organoid cultures were established from biopsies obtained during endoscopic or surgical procedures utilizing methods developed by the laboratory of Dr. Hans Clevers (*Sato et al., 2011*). De-identified biopsy tissue was obtained from healthy adult subjects who provided informed consent at Johns Hopkins University and all methods were carried out in accordance with approved guidelines and regulations. All experimental protocols were approved by the Johns Hopkins University Institutional Review Board (IRB #NA 00038329). Briefly, organoids generated from isolated intestinal crypts were grown embedded in Matrigel (Corning, USA) in the presence of expansion medium (EM) consisting of Advanced DMEM F12 supplemented with 50% v/v Wnt3a conditioned medium (produced by L-Wnt3a cell line, ATCC CRL-2647), 20% v/v R-spondin-1 conditioned medium (produced by HEK293T cell line stably expressing mouse R-spondin1; kindly provided by Dr. Calvin Kuo, Stanford University, Stanford, CA), 10% v/v Noggin conditioned medium (produced by HEK293T cell line stably expressing mouse Noggin), 10 mM HEPES, 0.2 mM GlutaMAX, 1x B27 supplement, 1x N2 supplement, 1 mM n-acetyl cysteine, 50 ng/ml human epidermal growth factor, 10 nM human [Leu15] -gastrin, 500 nM A83-01, 10 µM SB202190, 100 µg/ml primocin. EM was replaced every other day and supplemented with 10 µM CHIR99021 and 10 µM Y-27632 during the first 2 days after passaging. Organoids were passaged every 7 days and used for chip seeding between passage numbers 5 and 30.

## Duodenum Intestine-Chip

The design and fabrication of Organ-Chips used to develop the Duodenum Intestine-Chip was based on previously described protocols (*Huh et al., 2013*). The chip is made of a transparent, flexible polydimethylsiloxane (PDMS), an elastomeric polymer. The chip contains two parallel microchannels (a $1 \times 1$ mm epithelial channel and a $1 \times 0.2$ mm vascular channel) that are separated by a thin (50 µm), porous membrane (7 µm diameter pores with 40 µm spacing) coated with ECM (200 µg/ml collagen IV and 100 µg / ml Matrigel at the epithelial side and 200 µg / ml collagen IV and 30 µg / ml fibronectin at the vascular side). Chips were seeded with intestinal epithelial cells obtained from enzymatic dissociation of organoids, as described previously (*Kasendra et al., 2018*), and incubated overnight before being washed with fresh media. The next day, chips were connected to the culture module instrument (inside the incubator), that can hold up to 12 chips and allows for control of flow and stretching within the chips using pressure driven flow (*Vatine et al., 2019*). Chips were maintained under constant perfusion of fresh expansion medium at 30 µl/hr through top and bottom channels of all chips until day 6. Human Intestinal Microvascular Endothelial Cells (HIMECs; Cell Biologics) were than plated on the vascular side of the ECM-coated porous membrane in EGM-2MV medium, which contains human epidermal growth factor, hydrocortisone, vascular endothelial

growth factor, human fibroblastic growth factor-B, R3-Insulin-like Growth Factor-1, Ascorbic Acid and 5% fetal bovine serum (Lonza Cat. no. CC-3202). At this time, the medium supplying the epithelial channel was switched to differentiation medium. Differentiation medium consisted of the same components as those of Expansion Medium with 50% less Noggin and R-spondin-1 and devoid of Wnt3a and SB202190. Media supplying both chip channels were under continuous flow. Cyclic, peristalsis-like deformations of tissue attached to the membrane (10% strain; 0.2 Hz) were initiated after the formation of a confluent monolayer at ~4 days in culture.

## Permeability assays

In order to evaluate the establishment and integrity of the intestinal barrier, 3 kDa Dextran, Cascade Blue was added to the epithelial compartment of the Duodenum Intestine-Chip at 0.1 mg/ml on the day of their connection to flow. Effluents of the endothelial compartment were sampled every 48 hr to determine the concentration of dye that had diffused through the membrane. The apparent paracellular permeability (Papp) was calculated based on a standard curve and using the following formula:

$$Papp\left(\frac{cm}{s}\right) = \frac{C_{output}\left(\frac{mg}{ml}\right) x \, Flow \, rate \left(\frac{ml}{s}\right)}{C_{input}\left(\frac{mg}{ml}\right) x \, A \, (cm^2)}$$

where $C_{output}$ is the concentration of dextran in the effluents of the endothelial compartment, $A$ is the seeded area, and $C_{input}$ is the input concentration of dextran spiked into the epithelial compartment. The establishment of intestinal barrier function in Duodenum Intestine-Chip was evaluated in three independent experiments, each performed using chips established from a different donor of biopsy-derived organoids. At least three different chips were used per condition.

## Morphological analysis

Immunofluorescent staining of cells in the Duodenum Intestine-Chip was performed with minor modifications to previously reported protocols (*Kasendra et al., 2018*). Cells were fixed with 4% formaldehyde or cold methanol, and when required, were permeabilized, using 0.1% Triton X-100. 5% (v/v). Donkey serum solution in PBS was used for blocking. Incubation with primary antibodies directed against ZO-1, VE-cadherin, E-cadherin, villin, mucin 2, lysozyme, chromogranin A, MDR1, BCRP, PEPT1 (see Key Resources Table) was performed overnight at 4°C. Chips treated with corresponding Alexa Fluor secondary antibodies (Abcam) were incubated in the dark for 2 hr at room temperature. Cells were then counterstained with nuclear dye DAPI. Images were acquired with an inverted laser-scanning confocal microscope (Zeiss LSM 880 with Airyscan).

Chips processed for SEM were fixed in 2.5% glutaraldehyde, treated with 1% osmium tetroxide in 0.1 M sodium cacodylate buffer, dehydrated in a series of graded concentrations of ethanol solutions and critical point dried, as described previously (*Kasendra et al., 2018*). Prior to imaging, samples were coated with a thin (10 nm) layer of Pt/Pd using a sputter coater.

## Measurement of the density of microvilli

Images of microvilli on the surface of cells were captured with a scanning electron microscope (JSM-5600LV; JEOL). The morphological analysis and quantification of microvilli was performed using ImageJ. Number of intestinal microvilli per $\mu m^2$ were calculated after applying two image processing techniques, namely binarization and particle analysis, and Otsu's thresholding method, as described previously (*Julio et al., 2008*).

## MDR-1 efflux pump activity

The transporter activity of MDR-1 was assessed using the MDR1 Efflux Assay Kit, as per manufacturer's instructions. Briefly, Duodenum Intestine-Chip and Caco-2 Intestine-Chip (*Kim et al., 2012*; *Kim and Ingber, 2013*) were perfused apically with rhodamine 123, a fluorescent transport substrate of MDR1. Intracellular accumulation of dye was detected by fluorescent imaging (Olympus IX83) and measured in the presence and absence of MDR-1-specific inhibitor vinblastine (22 µM). Three independent experiments were performed for Caco-2 Intestine-Chip and Duodenum Intestine-Chip, each using chips established from a different donor of biopsy-derived organoids. At least three

different chips were used per condition. Images of each were taken at three different fields of view, digitally processed and quantified using Fiji software.

## CYP3A4 induction

Duodenum Intestine-Chip and Caco-2 Intestine-Chip were treated with 100 nM 1,25-dihydroxyvitamin D3 (Sigma) or 20 µM rifampicin (Sigma), which are known to induce CYP3A4, for 48 hr. Controls were treated with DMSO (final concentration 0.1%). CYP3A4 enzyme activity was determined using prototypical substrate testosterone (Sigma). Duodenum Intestine-Chip was incubated with 200 µM testosterone for 1 hr under a flow rate of 300 µL/hr. The reaction was stopped using acetonitrile with 0.1% formic acid, and formation of 6β-hydroxytestosterone was measured using LC-MS at In Vitro ADMET Laboratories, Inc (IVAL). Specific activity of CYP3A4 was determined by dividing the total metabolite formed by the incubation time and normalized to protein contents (pmol/min/mg protein). In order to assess the level of CYP3A4 induction at the gene and protein level cells in the epithelial channel were harvested either for RNA isolation and gene expression analysis or for western blotting, respectively.

## Western blotting

RIPA cell lysis buffer (Pierce) supplemented with protease and phosphatase inhibitors (Sigma) was used for the extraction of total protein from epithelial cells in the chips. The protein concentration in each sample was determined using the bicinchoninic acid method. Equal amounts (15 µg) of protein lysates were heat denatured and separated on a 4–10% Mini-Protean Precast Gel (Bio-Rad), followed by transfer on a nitrocellulose membrane (Bio-Rad). After blocking with 5% nonfat milk, membranes were probed with primary antibodies for CYP3A4 (mouse monoclonal, Santa Cruz Biotechnology) and GAPDH (rabbit polyclonal, Abcam) and incubated overnight at 4°C, followed by incubation for 1 hr with IRDye-conjugated secondary antibodies against rabbit and mouse immunoglobulin G (LI-COR), at room temperature. Finally, blots were scanned using an Odyssey Infrared Imaging System (LI-COR) and the protein bands were visualized and quantified using Image Studio software (LI-COR). Gluceraldehyde-3-phosphate dehydrogenase (GAPDH) was used as the loading control.

## Gene expression analysis

Total RNA was isolated from the epithelial cells grown in chip using PureLink RNA Mini kit (Thermo Fisher Scientific) and reverse transcribed to cDNA using SuperScript IV Synthesis System (Thermo Fisher Scientific). qRT-PCR was performed using TaqMan Fast Advanced Master Mix (Applied Biosystems) and TaqMan Gene Expression Assays (see *Supplementary file 2*, Thermo Fisher Scientific) in QuantStudio 5 PCR System (Thermo Fisher Scientific). Relative expression of gene was calculated using 2-ΔΔCt method.

## RNA isolation and sequencing

RNA was extracted using TRIzol (TRI reagent, Sigma) according to manufacturer's guidelines. Samples were submitted to GENEWIZ South Plainfield, NJ for next generation sequencing. After quality control and a complementary DNA library creation, all samples were sequenced using HiSeq 4000 with 2 × 150 bp paired-end reads per sample.

## RNA sequencing bioinformatics

Pre-processing: raw sequence data (.bcl files) generated from Illumina HiSeq was converted into fastq files and de-multiplexed using Illumina's bcl2fastq 2.17 software. Read quality was assessed using FastQC. Adaptor and low-quality (<15) sequences were removed using Trimmomatic v.0.36. The trimmed reads were mapped to the *Homo sapiens* reference genome available on ENSEMBL using the STAR aligner v.2.5.2b. The STAR aligner uses a splice aligner that detects splice junctions and incorporates them to help the alignment of the entire read sequences. BAM files were generated following this step. Unique gene raw counts were calculated by using feature Counts from the Subread package v.1.5.2. Only the unique reads that fell within exon regions were counted.

   Differential Gene Expression Analysis: to combine our gene expression dataset with publicly available data for adult duodenum gene expression (provided as Fragments Per Kilobase of transcript per Million mapped read (FPKM) values in *Finkbeiner et al., 2015*, we converted the raw counts to

FPKMs. Then, using the $\log_e$(FPKM) expressions of the combined datasets, we applied DE gene analysis using the R package 'limma'(*Ritchie et al., 2015*). For each comparison, the thresholds used to identify the DE genes were set to a) adjusted p-values<0.05 and b) absolute log2 fold change >2.

## GO term enrichment analysis and KEGG pathway analysis

After expression pattern clustering, the transcripts from specific groups were subjected to functional annotation, including GO (The Gene Ontology) functional annotation and KEGG (Kyoto Encyclopedia of Genes and Genomes) pathway annotation. The GO terms and KEGG pathway enrichment was performed using The Database for Annotation, Visualization and Integrated Discovery (DAVID v 6.8, http://david.abcc.ncifcrf.gov).

## Statistical analysis

All experiments were performed in triplicates and repeated with organoids from three different human donors. One-way or two-way ANOVA was performed to determine statistical significance, as indicated in the figure legends. The error bars represent standard error of the mean [s.e.m]; p values < 0.05 and above were considered as significant.

## Additional information

### Competing interests
Magdalena Kasendra, Raymond Luc, Dimitris V Manatakis, Gauri Kulkarni, Carolina Lucchesi, Josiah Sliz, Athanasia Apostolou, Jenifer Obrigewitch, Kyung-Jin Jang, Geraldine A Hamilton, Katia Karalis: is an employee and shareholder of Emulate. The other authors declare that no competing interests exist.

### Funding
No external funding was received for this work.

### Author contributions
Magdalena Kasendra, Conceptualization, Data curation, Formal analysis, Supervision, Validation, Investigation, Visualization, Methodology, Writing—original draft, Writing—review and editing; Raymond Luc, Formal analysis, Validation, Investigation, Methodology, Writing—original draft, Writing—review and editing; Jianyi Yin, Formal analysis, Investigation, Methodology, Writing—review and editing; Dimitris V Manatakis, Data curation, Formal analysis, Methodology, Writing—original draft, Writing—review and editing; Gauri Kulkarni, Carolina Lucchesi, Athanasia Apostolou, Laxmi Sunuwar, Jenifer Obrigewitch, Investigation, Writing—review and editing; Josiah Sliz, Formal analysis, Methodology; Kyung-Jin Jang, Methodology; Geraldine A Hamilton, Project administration, Writing—review and editing; Mark Donowitz, Conceptualization, Writing—review and editing; Katia Karalis, Conceptualization, Supervision, Writing—review and editing

### Author ORCIDs
Magdalena Kasendra (iD) https://orcid.org/0000-0002-6376-2489

### Ethics
Human subjects: De-identified biopsy tissue was obtained from healthy adult subjects who provided informed consent at Johns Hopkins University and all methods were carried out in accordance with approved guidelines and regulations. All experimental protocols were approved by the Johns Hopkins University Institutional Review Board (IRB #NA 00038329).

### Decision letter and Author response
Decision letter https://doi.org/10.7554/eLife.50135.sa1
Author response https://doi.org/10.7554/eLife.50135.sa2

# Additional files

## Supplementary files

- Supplementary file 1. RNAseq datasets downloaded from public databases.
- Supplementary file 2. List of human TaqMan gene expression assays used for qRT-PCR.
- Transparent reporting form

## Data availability

RNA sequencing data have been deposited in the National Center for Biotechnology Information Gene Expression Omnibus (GEO) under accession number GSE135196.

The following dataset was generated:

| Author(s) | Year | Dataset title | Dataset URL | Database and Identifier |
|---|---|---|---|---|
| Kasendra M, Luc R, Manatakis DV | 2019 | Genome-wide transcriptome profiling of human duodenal organoids, Duodenum Intestine-Chip and adult duodenal tissue using RNA-seq. | https://www.ncbi.nlm.nih.gov/geo/query/acc.cgi?acc=GSE135196 | NCBI Gene Expression Omnibus, GSE135196 |

The following previously published dataset was used:

| Author(s) | Year | Dataset title | Dataset URL | Database and Identifier |
|---|---|---|---|---|
| Bjorn M Hallstrom | 2013 | RNA-seq of coding RNA from tissue samples of 95 human individuals representing 27 different tissues in order to determine tissue-specificity of all protein-coding genes | https://www.ebi.ac.uk/arrayexpress/experiments/E-MTAB-1733 | EMBL-EBI ArrayExpress, E-MTAB-1733 |

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
