## [Decision Letter]

**Acceptance summary:**

In this manuscript, the authors extend the characterization of a human duodenum intestine-on-a-chip system, which they have described previously, and demonstrate its physiological relevance to real human duodenum tissue. The work will be interesting to a broad community of researchers interested in tissue biology and drug development.

**Decision letter after peer review:**

Thank you for submitting your article "Organoid-derived Duodenum Intestine-Chip for preclinical drug assessment in a human relevant system" for consideration by *eLife*. Your article has been reviewed by two peer reviewers, and the evaluation has been overseen by Anna Akhmanova as the Senior Editor. The reviewers have opted to remain anonymous.

The reviewers have discussed the reviews with one another and the Reviewing Editor has drafted this decision to help you prepare a revised submission.

Summary:

This paper provides a model of duodenum on a chip that uses human cells derived from organoids. The key element of the method appears to be the use of duodenum organoid fragments during the seeding process, which was applied in both the previous work and this current work. The physiologic mimicry of the duodenum intestine-chip proved to be excellent, and was superior to the organoids in their original form (not in a chip). Figure 3D is particularly impressive, with transcriptomic signatures between chip and duodenum tissue almost identical, and completely different to the signatures from non-chip organoids.

Essential revisions:

In many cases the authors conclude that there is no difference between native duodenum and duodenum on a chip, but this might just be due to a low N combined with relatively large data variability The reviewers suggest to provide power analysis to determine if significant differences have been missed. Rather than the new model being equivalent to native duodenum, a claim of being "closer to in vivo" than in the other conditions is much more appropriate.

While Figure 3D is impressive in many ways in terms of the underlying message, the authors need to address what appears to be some mislabeling and/or duplication of the data. Two of the heatmaps, for "metabolism of xenobiotics" and "chemical carcinogenesis" have exactly the same heatmap. Also, there are labels for genes in the "retinol metabolism" heatmap that are the same genes as those in the "metabolism of xenobiotics" heatmap, but have different results (for example, compare the heatmaps for CYP3A4, UGT2B17, UGT2B15). While it is possible for some genes to belong to more than one pathway, it is confounding that the heatmaps are entirely identical in one case, and then have differential expression patterns for a subset of overlapping genes in another case.

The authors used organoids for the differentiation of their cells and use cells derived from oranoids as a basis for their chips. The authors should rephrase it in the manuscript (e.g. to "organoid-derived cells") since the current wording leaves the impression that they integrate entire organoids.

The authors should clarify in more detail what discriminates this study from their previous one reported in Kasendra et al. Scientific Reports (2018) (DOI:10.1038/s41598-018-21201-7) and what the level of novelty is.

The gene expression data (Figure 2A) shows that the physiological "maturation" stops at day 8 and the chips then seems to degrade. Can the author explain this?

It is often unclear at what timepoints, the various endpoint analysis is performed. This is important information and should be included in each figure caption as well.

In general, the authors provide a very nice and comprehensive characterization of their model. Yet, they do not look at actual CYP3a4 activity, which would seem like an easy and straight forward assay that should be added. It is well known that amount of RNA and protein do not directly relate to amount of enzymatic activity.

---

## [Author Response]

Essential revisions:In many cases the authors conclude that there is no difference between native duodenum and duodenum on a chip, but this might just be due to a low N combined with relatively large data variability The reviewers suggest to provide power analysis to determine if significant differences have been missed. Rather than the new model being equivalent to native duodenum, a claim of being "closer to in vivo" than in the other conditions is much more appropriate.

We agree with the reviewers that the claim that Duodenum Intestine-Chip provides a “closer to in vivo” model of the human duodenum, compared to organoids or Caco-2 cells, reflects well the current state of this system. Indeed, in the first version of the manuscript we often used the above or similar term (e.g. “closer to *in-vivo*, … as compared to, “closer to human duodenal tissue than” etc.) in order to describe with accuracy, the system that we have developed. Please find specific examples below

- closer to in vivo, …as compared to Caco2 cells (Introduction section)

- closer similarity … to human duodenum tissue in comparison to organoids (subsection “Transcriptomic comparison of the Duodenum Intestine-Chip versus organoids”)

- closer emulation of human in vivo tissue than … (subsection “Intestinal drug transporters and MDR1 efflux activity in the Duodenum Intestine-Chip”)

- closer to human duodenal tissue than … (subsection “Drug-mediated CYP3A4 induction in the Duodenum Intestine-Chip”)

- closer to human duodenal tissue … in comparison to … (Discussion paragraph five)

- increased similarity … between Duodenum Intestine-Chip and human adult duodenum tissue in comparison to … (subsection “Transcriptomic comparison of the Duodenum Intestine-Chip versus organoids”)

Importantly, we have never referred to Duodenum Intestine-Chip as being “equivalent” or “equal” to native duodenum. Moreover, we would like to highlight that in the first and revised version of the manuscript we have reported the exact differences identified, using RNA sequencing, between Duodenum Intestine-Chip and adult duodenal tissue, including the number of genes that differ in between the two and the biological processes associated with the identified gene set (Figure 3—figure supplement 1B). Following reviewers comment, we have revised the manuscript thoroughly to clarify our claims and remove any unintentional overstatements.

Additionally, in order to address the concerns raised by the reviewers in respect to the small sample size of the present study, we now include a more detailed description of the strict statistical methods used for the RNA-seq data analysis, which strongly supports the validity of the presented results. More specifically, for the DGE analyses we used the limma R-package (Ritchie et al., 2015) with a proven excellent performance at low sample numbers and strict thresholds (i.e. adjusted p-value < 0.05 and |log fold change| > 2) for the selections of differentially expressed genes. Additionally, in all statistical hypothesis testing performed (in DGE and pathway analyses) we used as a threshold an adjusted p-value (instead of the p-value), which reduces the false discovery rate and increases the statistical power of the findings.

While Figure 3D is impressive in many ways in terms of the underlying message, the authors need to address what appears to be some mislabeling and/or duplication of the data. Two of the heatmaps, for "metabolism of xenobiotics" and "chemical carcinogenesis" have exactly the same heatmap. Also, there are labels for genes in the "retinol metabolism" heatmap that are the same genes as those in the "metabolism of xenobiotics" heatmap, but have different results (for example, compare the heatmaps for CYP3A4, UGT2B17, UGT2B15). While it is possible for some genes to belong to more than one pathway, it is confounding that the heatmaps are entirely identical in one case, and then have differential expression patterns for a subset of overlapping genes in another case.

We thank the reviewers for their careful reading of the manuscript and their constructive remarks which helped us to improve the quality of the manuscript. As correctly pointed out by the reviewers, a subset of overlapping genes identified through DGE showed to belong to two different KEGG pathways “metabolism of xenobiotics by cytochrome P450” and “chemical carcinogenesis” resulting in the generation of two identical heatmaps (Figure 3D). Since these pathways belong to two different categories in KEGG Orthology (please refer to https://www.genome.jp/kegg/pathway.html): “Metabolism” and “Human diseases”, we have included both heatmaps in the first version of the manuscript. Based on the feedback received from the reviewers we decided to combine these results and present them in a clear and concise format (Figure 3D).

We thank the reviewers for pointing out the “different expression patterns for a subset of overlapping genes” (e.g. *CYP3A4, UGT2B17, UGT2B15*) across different heatmaps (e.g. "retinol metabolism" and “metabolism of xenobiotics”) that helped us to better explain the results and improve the figure caption to avoid confusion from the readers. Below we provide a brief clarification and description of the changes introduced to the manuscript.

In order to generate Figure 3D we have used the R-function “pheatmap”, which automatically selects the optimal range of values for creating the color scale of each heatmap ensuring construction of high-quality graphics. The differences in the automatically selected range of log_e_(FPKM) values used to create the color scale of each heatmap explains the observed differences in the colors representing the expression of the same gene in between different graphs. For example, comparing the expression levels of UGT2B17 in the “Metabolism of xenobiotics by cytochrome P450” (see the color bar on the left site of the heatmap) we can see that it’s expression in organoids ranges between log_e_(FPKM) 0.73 to 2.20, similarly to its expression in “metabolism of retinol” (see the color bar on the left site of the heatmap), although they are represented by different colors ranging between light and dark blue or white to light blue, respectively. In the revised manuscript we clarify that different range of log_e_(FPKM) values and corresponding to them color scales have been used for generation of different heatmaps.

The authors used organoids for the differentiation of their cells and use cells derived from oranoids as a basis for their chips. The authors should rephrase it in the manuscript (e.g. to "organoid-derived cells") since the current wording leaves the impression that they integrate entire organoids.

In the revised manuscript we use term ‘organoid-derived cells’, to make a clear distinction from the intact/entire organoids, as correctly suggested by the reviewers.

The authors should clarify in more detail what discriminates this study from their previous one reported in Kasendra et al. Scientific Reports (2018) (DOI:10.1038/s41598-018-21201-7) and what the level of novelty is.

We appreciate the reviewers’ constructive suggestion which helped to clarify the goal and novelty of our new work submitted for your consideration. While in our previous proof-of-concept study (Kasendra et al., 2018), we demonstrated that biopsy-derived intestinal organoids and Organs-on-Chips are complementary technologies which, when combined enable the development of a physiologically relevant model of human small intestine. In this study, we provide a comprehensive functional and phenotypic characterization of this model to show the clear advantages for the synergistic use of the two technologies and we demonstrate how it can be applied for drug discovery and development applications. In more details, we were able to demonstrate that the physiological mimicry of the Duodenum Intestine-Chip is excellent and superior to the organoids in their original form (not in a chip). We showed the critical contribution of mechanical forces emulating intestinal blood flow (fluid flow) and peristalsis (stretch) for the establishment of correct intestinal tissue cytoarchitecture (polarization, junction integrity, presence of intestinal microvilli), which underlie its main biological functions such as nutrient absorption, digestion and transport, drug transport and metabolism. Using RNA sequencing approach, we identified a subset of genes and associated with them biological pathways/processes which are responsible for the closer, compared to organoids, resemblance of Intestine-Chip system to human duodenal tissue. Moreover, we demonstrated an improved, as compared to Caco-2 cells or organoids, expression of cytochrome P450 (CYP) 3A4 in Duodenum Intestine-Chip, which along with the successful modulation of its expression by the prototypical inducers (rifampicin and 1,25-dihydroxyvitamin D3) makes its a better suited model for studying intestinal first-pass metabolism and drug-drug interactions.

Following the reviewer's suggestion, we have clarified in the Discussion section of the revised manuscript what discriminates this work from our previous study, both in terms of the methodology used as well as novel findings.

The gene expression data (Figure 2A) shows that the physiological "maturation" stops at day 8 and the chips then seems to degrade. Can the author explain this?

The expression of markers specific for intestinal cell types that naturally reside in the villus compartment – alkaline phosphatase of absorptive enterocytes, mucin 2 for goblet cells, chromogranin A for enteroendocrine cells – showed a linear increase up to day 8 of Duodenum Intestine-Chip culture. These findings suggest the progress in epithelial cell differentiation over the time of culture on-chip. The withdrawal of the growth factors (Wnt3A, EGF) from the culture media, is followed by the physiological shedding, that together with the lack of a renewable stem cell niche accounts for the gradual “degradation” of the chip after day 8-10. In other words, with given culture conditions, the expansion of organoid-derived cells toward differentiated, polarized intestinal epithelial monolayers come at the expense of “indefinite” maintenance of stem cells. Thus, limiting the capability of the system to model the perpetual self-renewal.

It is often unclear at what timepoints, the various endpoint analysis is performed. This is important information and should be included in each figure caption as well.

We updated the manuscript to include all of the important missing information pointed out by the reviewer, both in the main text as well as in each figure caption.

In general, the authors provide a very nice and comprehensive characterization of their model. Yet, they do not look at actual CYP3a4 activity, which would seem like an easy and straight forward assay that should be added. It is well known that amount of RNA and protein do not directly relate to amount of enzymatic activity.

Following the reviewers’ suggestion, in the revised manuscript we included the data demonstrating baseline and induced CYP3A4 activity in Duodenum and Caco-2 Intestine-Chip (Figure 4D).